# Astrocyte Immune Functions and Glaucoma

**DOI:** 10.3390/ijms24032747

**Published:** 2023-02-01

**Authors:** Youichi Shinozaki, Kenji Kashiwagi, Schuichi Koizumi

**Affiliations:** 1Department of Neuropharmacology, Interdisciplinary Graduate School of Medicine, University of Yamanashi, Yamanashi 409-3898, Japan; 2Interdisciplinary Brain-Immune Research Center, Interdisciplinary Graduate School of Medicine, University of Yamanashi, Yamanashi 409-3898, Japan; 3Department of Ophthalmology, Interdisciplinary Graduate School of Medicine, University of Yamanashi, Yamanashi 409-3898, Japan

**Keywords:** astrocytes, glaucoma, immune function, inflammation, Müller cells, ONH, retina, RGCs

## Abstract

Astrocytes, a non-neuronal glial cell type in the nervous system, are essential for regulating physiological functions of the central nervous system. In various injuries and diseases of the central nervous system, astrocytes often change their phenotypes into neurotoxic ones that participate in pro-inflammatory responses (hereafter referred to as “immune functions”). Such astrocytic immune functions are not only limited to brain diseases but are also found in ocular neurodegenerative diseases such as glaucoma, a retinal neurodegenerative disease that is the leading cause of blindness worldwide. The eye has two astrocyte-lineage cells: astrocytes and Müller cells. They maintain the physiological environment of the retina and optic nerve, thereby controlling visual function. Dysfunction of astrocyte-lineage cells may be involved in the onset and progression of glaucoma. These cells become reactive in glaucoma patients, and animal studies have suggested that their immune responses may be linked to glaucoma-related events: tissue remodeling, neuronal death, and infiltration of peripheral immune cells. In this review, we discuss the role of the immune functions of astrocyte-lineage cells in the pathogenesis of glaucoma.

## 1. Introduction

Astrocytes are a subtype of glial cells in the central nervous system. Glial cells are non-neuronal cell types with essential roles in regulating physiological brain functions and are divided into several types, including oligodendrocytes, microglia, and astrocytes [1]. They are not only limited to the brain but are also distributed in the peripheral nervous system, including the eye. The retina has two types of astrocyte-lineage cells: astrocytes and retina-specific Müller cells. Astrocytes are highly plastic cells and change their phenotypes into either neurodegenerative or neuroprotective ones under pathological conditions [2,3]. Astrocytes respond to inflammatory signals and can trigger inflammation by changing their phenotype into a neurodegenerative one. The neurodegenerative astrocytes are an important driver of various neurological diseases, including Alzheimer’s disease [4], Parkinson’s disease [5], Huntington’s disease [6], amyotrophic lateral sclerosis [7], and multiple sclerosis [8]. Various factors induce neurotoxic astrocytes, including lipopolysaccharide [9], normal brain aging [10], human apoE4 knock-in [11], activated endothelial cells [12], and microglia-derived fragmented mitochondria [13]. However, other factors/conditions induce neuroprotective and anti-inflammatory astrocytes, including reduction in P2Y_1_ receptor expression [14], ischemia-reperfusion [15], spinal cord injury [16], and type I interferons and microbial metabolite tryptophan [17,18]. These reports suggest that astrocytic neurodegenerative inflammatory responses or neuroprotective anti-inflammatory responses can be controlled by various factors, and that the inflammation caused by astrocytes would be a novel therapeutic target for neurodegenerative diseases. In this review, we discuss the pathogenic roles of astrocytes and Müller cells in glaucoma, especially focusing on their pro-inflammatory responses (hereafter referred to as “immune responses”).

## 2. Glaucoma

Glaucoma, a progressive optic neuropathy that affects more than 70 million people worldwide [19,20], is the second most common cause of blindness worldwide. In glaucoma, blindness is caused by damage to the optic nerve and degeneration of retinal ganglion cells (RGCs), retinal neurons essential for transducing visual information to the brain. Elevated intraocular pressure (IOP) is one of the most well-known risk factors for glaucoma, and the simplest interpretation is that IOP-mediated stress damages RGCs (Figure 1). Although reducing IOP can prevent or delay vision loss [21,22], IOP elevation alone does not determine whether patients develop glaucoma, and many patients develop glaucoma regardless of their IOP level [22]. In addition, some individuals show ocular hypertension, a diagnosis applied when the IOP is above the normal range, without any optic nerve damage [23]. Importantly, a substantial percentage of glaucoma patients (approximately 30%–40% in Caucasians) show normal IOP levels, i.e., normal-tension glaucoma (NTG) [24]. The Asian, and especially Japanese [25], population [26] shows a significantly higher prevalence of NTG (52%–92%). These reports suggest that risk factors other than IOP are also important for the pathogenesis of glaucoma.

 Glaucomatous ocular tissue damage includes anatomical and functional deterioration of the retina and optic nerve head (ONH). The optic disc in the ONH is the location of the retina at which RGC axons gather together, form an optic nerve, and exit the eye [27,28]. The optic disc under ophthalmoscopy is observed as a blight circle near the center part of the eye. There is a bright spot at the center of the optic disc, namely the ‘(optic) cup’. Enlargement of ONH excavation (referred to as ‘cupping’, i.e., narrowing of the neuronal rim and an increase in the cup-to-disk ratio) is the hallmark of anatomical changes in human patients with hypertensive glaucoma (primary open-angle glaucoma, POAG) and NTG [27] (Figure 2). The ONH has been suggested to be the primary site of injury in glaucoma. The primate glaucoma model and post-mortem tissue of glaucoma patients revealed that RGC axons are lost at the lamina cribrosa [29,30]. The lamina cribrosa is an extracellular matrix (ECM)-rich (mainly collagen) structure, located just behind the optic cup, which provides physical support to the axon fibers as they pass through the posterior wall of the eye. This collagen-rich structure (i.e., ECM plate) is limited in the lamina cribrosa of humans and non-human primates. In rodents, since the region corresponding to the human LC lacks the ECM plate but astrocytes are abundant, this region is referred to as the “glial lamina” [31]. Optic nerve atrophy at the ONH precedes visual impairment in glaucoma [32], and Wallerian degeneration-like mechanisms may cause RGC soma death [33]. In addition to the RGC soma loss, thinning of the retinal nerve fiber layer, a well-known tissue change found in glaucoma patients, is associated with visual field loss [27]. Dendritic atrophy is also a characteristic feature of RGCs in both glaucoma patients [34] and model animals [35]. Such RGC damage and death induce visual impairment in glaucoma.

Elevated IOP is a well-known risk factor for hypertensive glaucoma (POAG). The simplest hypothesis is that IOP-mediated physical stress may directly damage retinal ganglion cells (RGCs) and the optic nerve. Accumulating evidence suggests that elevated IOP alone does not determine whether patients develop glaucoma but that other factors also contribute to the pathogenesis of glaucoma. In the case of NTG, glaucoma is developed even if the patients have a normal level of IOP.

In glaucoma patients, a central depression of the retina (i.e., the optic cup) is deformed, leading to posterior displacement and enlargement of the optic cup. The lamina cribrosa, a collagen-enriched tissue, provides physical support to the retinal ganglion cell (RGC) axons. With the enlargement of the optic cup, the structural remodeling of the lamina cribrosa physically stresses and damages the optic nerve. The normal and neuroprotective astrocytes support RGC axon integrity by releasing neurotrophic factors and/or anti-inflammatory molecules. Under glaucoma, astrocytes release neurotoxic factors such as cytokines and chemokines and damage RGC axons.

## 3. Astrocytes, Immune Responses, and Pathogenesis of Glaucoma

The retina, ONH, and optic nerve contain several glial types, such as astrocytes, Müller cells, and microglia (Figure 3). Astrocytes localize at the inner surface of the retina and throughout the optic nerve. Müller cells exist only in the retina and vertically span the entire thickness of the retina. Oligodendrocytes exist in the distal part of the optic nerve. GFAP-positive astrocytes have been shown to be closely associated with Tuj1-positive RGC axons in mouse retinal slices (Figure 4a). Visualization of Müller cells by YC-Nano under the control of Mlc-tTA [36] revealed that they extend their processes toward and attach to SMI32-positive RGC somas (asterisk, Figure 4b). Some Müller cell end feet enwrap RGC axons (arrow, Figure 4b). At the ONH, astrocytes are accumulated along with RGC axons (Figure 4c). Horizontal slices of ONH show that GFAP-positive astrocytes form honeycomb structures, and that SMI32-positive RGC axons pass through them (Figure 4d). Similar to those found in the brain, ocular astrocyte-lineage cells help maintain homeostatic retinal functions, including metabolic support and nutrition of neurons, ion buffering, water transport, neurotransmitter uptake, and blood vessel regulation, and they can act as optical fibers [37,38,39]. 

Inflammation may be related to the pathogenesis of glaucoma [40,41]. The complement pathway, essential for immune responses and inflammation, may also be related to the pathogenesis of glaucoma. Retinal astrocytes up-regulate complement C1q expression in RGCs [42]. The DBA/2J mouse, an inherited glaucoma model that develops elevated IOP and RGC death, shows up-regulated retinal C1q expression. Because C1q and its downstream C3 signaling mediate synapse elimination, overactivation of these signaling pathways may be an initial cause of the pathogenesis of glaucoma. Recent single-cell RNA-sequence analysis revealed that Müller cells are the major contributor to complement activations in the retina [43]. Aging, an important risk factor for glaucoma, increases various complement expressions in the retina. Transient ocular ischemia also up-regulates multiple complement molecules; this is not limited to rodents, but is also observed in the non-human primate model of glaucoma and human glaucoma patients [44]. These reports suggest that both astrocytes and Müller cells may contribute to evoking immune responses and the pathogenesis of glaucoma. 

A schematic showing glial cells in the retina. In addition to retinal neurons (e.g., RGCs, ACs, BPs, HCs, and rod and cone photoreceptors), there are several types of glial cells in the retina. Astrocytes are located at the inner surface of the retina. Müller cells vertically span the entire thickness of the retina. Microglia exist in various neuronal layers, such as the NFL, GCL, IPL, and OPL. NFL: nerve fiber layer; GCL: ganglion cell layer; IPL: inner plexiform layer; INL: inner nuclear layer; OPL: outer plexiform layer; ONL: outer nuclear layer; IS/OS: inner or outer segment of photoreceptors; RGC: retinal ganglion cell; AC: amacrine cell; BP: bipolar cell; HC: horizontal cell.

Extracellular ATP may also regulate immune responses by astrocyte-lineage cells. Extracellular ATP binds to receptors in the plasma membrane, namely purinergic P2 receptors. ATP-mediated purinergic signaling is an important regulator for damage-associated molecular patterns or alarmins [45]. ATP is released or leaked from various cells and tissues in the eye [46]. Under physiological conditions, extracellular ATP levels are tightly regulated and maintained at very low concentrations. Under pathological conditions, extracellular ATP levels are often elevated by leakage and/or release from damaged cells or other cell types. High ATP concentrations activate P2X7 receptors, a high-threshold type of P2 receptor, and trigger pro-inflammatory responses via astrocyte-lineage cells. Activation of astrocyte P2X7 receptors induces the production of chemokines, cytokines, and reactive oxygen species, and the priming of the nucleotide-binding and oligomerization domain-like receptor family pyrin domain-containing protein 3, an essential inflammasome regulator [47,48,49]. P2X7 receptor activation also induces cytokine production in Müller cells [50]. Astrocyte-mediated immune responses may regulate neuronal functions and damage neurons via controlling microglia or peripheral immune cells [51,52,53,54]. Immune cell responses may regulate microglia-to-astrocyte signaling and neurotoxicity. Microglia can release ATP by exocytosis [55], and activation of microglial TLR4 triggers ATP release and P2Y_1_ receptor activation in astrocytes, which induces glutamate release and enhances excitatory neurotransmission [56]. Another study has shown that microglia-derived ATP activates the astrocytic P2Y_1_ receptor and induces interleukin (IL)-6 expression, thereby inducing a neuroprotective effect [57] by blocking excitatory neurotransmission [58]. Astrocytic or microglial CXCR4 signaling enhances astrocytic glutamate release and causes neurotoxicity [59]. A recent study has suggested that C-X-C motif chemokine ligand 12 (CXCL12), an endogenous agonist for CXCR4, is up-regulated in retinal astrocytes in an NTG mouse model [60]. 

Previous reports have suggested that microglia dynamically regulate neuroprotective [14] or neurotoxic [9] astrocytic phenotypes. Neurotoxic astrocytes are induced by tumor necrosis factor (TNF)/IL-1α/C1q signals from microglia, and blocking this signaling protects RGCs from optic nerve crush damage. TNFα is also released from astrocytes, and astrocyte-derived TNFα enhances glutamate-mediated excitatory neurotransmission [61]. Although the hypothesis of glutamate-mediated RGC excitotoxicity in glaucoma patients is still under debate [62], many animal models have suggested that excitotoxicity is an attractive candidate for glaucoma. A recent study has proposed that astrocytic immune responses may cause excitotoxicity of RGCs under glaucomatous conditions [60]. A novel NTG mouse model showed that astrocytic immune responses were associated with a reduction in *Grin3a* expression in RGCs. NR3A, a dominant-negative subunit of the *Grin3a*-encoding *N*-methyl-D-aspartic acid (NMDA) receptor [63], suppresses Ca^2+^ permeability through the NMDA channel. In the novel NTG mouse model, certain RGC subgroups expressed high *Grin3a* expression, which was dramatically decreased in glaucoma [60]. Grin3a deficiency causes Ca^2+^ overload and excitotoxicity in RGCs [63,64]. The immune functions of astrocyte-lineage cells may be involved in glutamate-mediated RGC toxicity in glaucoma.

## 4. Autoimmunity and Glaucoma

Over the past decades, accumulating evidence from clinical and basic studies has suggested that dysregulated immune systems and autoimmunity may trigger the pathogenesis of glaucoma. Astrocyte-lineage cells may bridge autoimmunity and glaucoma. Abnormal immune activity can be observed [65] in glaucoma patients, of which many show an elevated prevalence of monoclonal gammopathy [66]. Glaucoma patients also show elevated levels of autoantibodies against crystallin, S100, and heat shock proteins (HSPs) in tears, serum [67,68,69], and aqueous humor [70]. 

The gene encoding αB-crystallin (*CRYAB*) is highly expressed in astrocytes and Müller cells in the human retina [71]. CRYAB was initially identified as a negative regulator of T-cell responses in experimental autoimmune encephalomyelitis [72], whereas phosphorylation of intracellular CRYAB in astrocytes mediates reactive astrogliosis and worsens the pathology of multiple sclerosis in both the mouse and human brain [73]. In the mouse model of glaucoma, the protein expression of crystallin decreased with age, and recombinant crystallin proteins protected RGCs in retinal explants [74]. Although CRYAB in astrocytes plays detrimental roles in the brain, reduced CRYAB levels may be linked to higher T-cell responses. An animal model has proposed that T cells play an active role in the pathogenesis of glaucoma [75]. 

To reproduce glaucoma autoimmunity, researchers employed immunization of animals by ocular tissue antigen homogenate and found that the immunized animals showed increased levels of autoreactive antibodies against ocular tissue [76]. The immunized animals also exhibited glaucoma-like tissue changes, such as RGC death, optic nerve atrophy, and reduced ocular responses without elevated IOP [77]. In addition to the tissue homogenates, S100β and HSPs were used for immunization. S100β, a Ca^2+^-binding protein, is highly expressed in astrocytes and Müller cells, which release S100β in response to various stimuli such as neuronal activity and high glucose [78,79]. When S100β protein is intravitreally administered, RGCs degenerate without changes to IOP [80]. S100β increases the expression of NFκB and complement C3 [81] and triggers microglial pro-inflammatory responses [82,83]. The S100β-evoked RGC damage and impaired ocular responses are recovered by minocycline [84], indicating the role of microglia in triggering glaucoma-like phenotypes. Whether S100β immunization induces T-cell responses is currently unclear. Autoimmunity against HSPs may also be related to glaucoma, and serum anti-HSPs antibodies were elevated in serum [67,68,69] and aqueous humor [70] of glaucoma patients. Because ONH astrocytes express and up-regulate Hsp27 in glaucoma patients [85], they might release HSPs and participate in autoimmune responses and RGC damage. In the case of HSPs, T cells infiltrate and damage RGCs by releasing Fas-ligand [86]. 

## 5. Glaucoma Risk Genes and Astrocyte-Lineage Cells

A recent study revealed that astrocyte-derived neurotoxic molecules include long saturated fatty acids and ELOVL fatty acid elongase 1 (ELOVL1) [87]. Previous genome-wide association studies have suggested that single-nucleotide polymorphisms (SNPs) of the *ELOVL5* gene correlate with a higher risk for NTG [88] and primary open-angle glaucoma [89]. Notably, the human protein atlas database (https://www.proteinatlas.org/, accessed on 28 January 2023) and previous single-cell transcriptome data from human retina have revealed that the *ELOVL5* gene is highly enriched in Müller cells rather than astrocytes [71,90,91]. These reports suggest lipid signaling also contributes to immune responses and the pathogenesis of glaucoma via glial cells.

Other immune response-related genes may also participate in the pathogenesis of glaucoma. The SNPs of the genes encoding Toll-like receptor-2 (*TLR2*) and -4 (*TLR4*), key molecules for anti-infectious and inflammatory responses, also correlate with a higher risk for NTG [92,93,94]. Although TLRs are highly enriched in the microglia of the brain, human retinal astrocytes highly express the *TLR4* gene [71]. The SNPs of the gene encoding Lysyl oxidase-like 1 (*LOXL1*), related to inflammation and fibrosis in the liver and lungs [95,96], also correlate with elevated risk for NTG [97,98]. Recent single-cell transcriptome studies have revealed that the *LOXL1* gene is abundant in Müller cells and astrocytes in humans [71]. Caveolin genes are also related to an elevated risk for NTG [99]. Both caveolin-1 and -2 genes (*CAV1* and *CAV2*) are highly enriched in astrocytes and Müller cells in the human retina [71]. *Cav1* knockout reduces lipopolysaccharide-evoked production of pro-inflammatory cytokines (CCL2, CXCL1, IL-6, and IL-1β) but increases the infiltration of leukocytes [100]. *Cav1* deficiency also reduces IL-6 receptor-mediated STAT3 activation and protects retinal neurons against sodium iodate [101], indicating that aberrant astrocyte/Müller cell immune responses may be involved in glaucomatous neurodegeneration. 

Glial cell senescence may be linked to the pathogenesis of glaucoma. The SIX homeobox 6 (*SIX6*) gene, a senescence-related gene, is also a risk gene for glaucoma, including NTG [102,103]. The *SIX6* gene is highly expressed in astrocytes and Müller cells in the human retina [71,90,91]. A previous study has demonstrated that SIX6 induces the expression of the senescence gene *p16^INK4^* and mediates the pathogenesis of glaucoma [104]. In the neurodegenerative disease mouse model, senescent astrocytes and microglia-protected neurons were depleted [105]. Because senescent cells release a group of soluble molecules known as senescence-associated secretory phenotype factors (e.g., chemokines, cytokines, and growth factors), immune responses may participate in astrocyte senescence-mediated neuronal damage. Gene variation of SIX homeobox 1 (*SIX1*) is also related to a higher risk for glaucoma [103,106]. SIX1 also controls senescence via p16^INK4^ [107] and is exclusively expressed in Müller cells [71]. Because SIX1 is an integral component of the non-canonical component of NFκB activation and suppresses inflammation [108], a functional deficit of SIX1 in Müller cells could induce their inflammatory responses. 

Intracellular Ca^2+^ is a well-known essential signal for regulating astrocyte functions [109,110]. Transmembrane and coiled-coil domains 1 is a Ca^2+^-load-activated Ca^2+^ channel that is expressed in and regulates Ca^2+^ levels within the endoplasmic reticulum [111]; its deficiency results in Ca^2+^ overflow into the endoplasmic reticulum [112]. Aberrant Ca^2+^ signaling in astrocytes is linked to neurological disease [113] and is often associated with pro-inflammatory responses. 

Connexin 43 encoded by the *Gja1* gene also has pivotal roles in maintaining the ocular microenvironment, and a recent study has suggested that *GJA1* gene variation is associated with an elevated risk for glaucoma [114] because astrocytic connexin is essential for gap junction formation. The astrocytic gap junction contributes to buffering of K^+^ ions and glutamate incorporated from the synaptic cleft. Dysfunction in gap junction formation limits astrocyte-mediated clearance of K^+^ and glutamate and causes neuronal hyperexcitability, which may be related to neurodegenerative diseases [115]. Associated with excitotoxicity, deficiency in astrocyte Cx43 exacerbates inflammation after brain injury [116]. In human patients with glaucoma, *GJA1* expression is higher than in age-matched healthy volunteers [117]. Because the Cx43 spatial pattern is changed in glaucoma [118], the gap junction may be lost under pathological conditions. High hydrostatic pressure causes internalization of Cx43 in human ONH astrocytes, indicating that the gap junction decouples intercellular communication under glaucomatous conditions [119]. The decoupled Cx43 (i.e., hemichannel) mediates ATP release and triggers inflammation. These reports suggest that many glaucoma-risk genes are expressed in astrocyte-lineage cells, and their dysfunctions may cause the pathogenesis and progression of glaucoma.

## 6. Astrocyte Responses in Glaucoma Patients 

In glaucoma patients, ocular astrocyte-lineage cells show dynamic structural and molecular changes [30,120,121,122,123,124,125]. Similar to other species, human astrocytes are also present at the inner surface of the retina along with RGC axons. They intimately associate with RGC axons and are highly accumulated at the ONH and optic nerve. As described above, the lamina cribrosa, a mesh-like structure that allows axons of the optic nerve to pass through the sclera in the ONH, is the primary site of damage in RGC axons [126,127]. The intraorbital and retrolaminar optic nerve is unmyelinated and thus directly surrounded by astrocytes [128], indicating that dysfunction of astrocytes at this site is critical for the pathogenesis of glaucoma. 

A previous study has shown that ONH astrocytes become reactive in glaucoma patients [126]. In an animal model of glaucoma, such changes are the earliest detectable event that is often observed before axonal damage [129]. As described above, ONH cupping is essential for axonal damage (Figure 2) and is induced by the reorganization of existing tissue via tissue destruction and ECM production. ONH astrocytes from glaucoma patients also express higher levels of matrix metalloproteinases (MMPs) [130] and ECM molecules [123]. Transforming growth factor β (TGFβ) mediates this process and is expressed in ONH astrocytes [131,132]. TGFβ levels in the ONH and aqueous humor are significantly higher in glaucoma patients [131]. TGFβ increases the expression levels of ECM and ECM-degrading enzymes, including MMPs [133]. Statin, a potent inhibitor of cholesterol synthesis, suppresses TGFβ-mediated MMP expression in human ONH astrocytes [134]. A recent study has suggested that deficiency in astrocytic ATP-binding cassette transporter A1, which transports cholesterol and fatty acids to the ECM [135], induces astrocytic pro-inflammatory responses and NTG-like pathology in mice [60]. Abnormal lipid signaling in astrocytes may accelerate tissue remodeling of the ONH, and a previous study has suggested that oxidative stress is involved in TGFβ-mediated ECM production in human ONH astrocytes [136]. Purinergic P2Y_1_ receptor signaling up-regulates antioxidant genes in astrocytes [137,138] and thus may be an attractive target for preventing ONH remodeling. In support of this hypothesis, P2Y_1_ receptor deficiency causes age-associated RGC damage [139]. Dysregulated extracellular ATP-mediated glial signaling can cause oxidative stress and damage RGCs in glaucoma [46]. 

In addition to tissue remodeling, inflammatory responses by astrocytes may be important for RGC damage. Human ONH astrocytes express nitric oxide synthase and up-regulate its expression in glaucoma [140,141], which may damage RGC axons [142,143]. ONH astrocytes from NTG patients also express TNFα [130]. In addition to astrocytes, Müller cells may contribute to RGC degeneration in glaucoma patients [144]. Up-regulated TNFα expression was observed in Müller cells in the retina of glaucoma patients [145]. Advanced glycation end-products are increased in glaucoma patients, and their receptors, essential regulators of inflammation [146], are up-regulated mainly in Müller cells in glaucoma patients [147]. Mice with advanced glycation end-product receptor knockout show a protective effect against ocular hypertension in RGCs [148], indicating that the inflammatory responses by Müller cells may cause RGC damage. At the perivascular region of the retina, reactive changes in astrocytes and Müller cells are observed in the early stages of glaucoma [149], and the retinal area at which astrocyte-lineage cells become reactive shows a significantly thinner retinal nerve fiber layer [150]. These studies suggest that immune responses of astrocyte-lineage cells are essential components of glaucoma pathogenesis.

Another molecule related to glaucoma is endothelin (ET), a potent vasoconstrictor. Plasma ET-1 levels are significantly elevated in NTG patients [151,152]. As intravitreal administration of ET-1 decreases IOP [151], ET-1-mediated RGC damage seems irrelevant for IOP elevation. Higher plasma ET-1 levels and SNPs of the genes encoding ET and ET receptors (ETRs) correlate with a higher risk for NTG [153,154,155,156]. Abnormal ET signaling could be involved in altered ocular circulation. Impaired ocular blood flow and disc hemorrhage are involved in the onset and progression of NTG [157,158,159]. In addition, immune responses by astrocyte-lineage cells contribute to the pathogenic roles of ET. The single-cell RNA-sequence database from the human retina has shown that ET and ETR are highly expressed in astrocytes and Müller cells [71,90,91]. Stimulation of human ONH astrocytes with ET-1 triggers [Ca^2+^]i transients and cell proliferation, which are mediated by ET_A_R and ET_B_R [160]. ET-2 induces reactive changes in Müller cells, disruption of the blood-retina barrier, and infiltration of peripheral immune cells [161]. The blockade of peripheral immune cells and treatment with bosentan, an ETR antagonist, show a protective effect on RGCs in DBA/2J mice [162,163]. ET-1 also increases the expression of MMP, tissue inhibitor metalloproteinase, and ECM in ONH astrocytes [164]. Taken together, immune responses by astrocyte-lineage cells are involved in the onset and progression of glaucoma.

## 7. Astrocytes as a Therapeutic Target for Glaucoma

As described above, previous studies have suggested that abnormal astrocytes trigger and accelerate the pathology of glaucoma. The prevention of long saturated fatty acids by knocking ELOVL1 out of astrocytes protected RGCs against optic nerve crush [87]. In the case of astrocytic ABCA1, the loss of ABCA1 causes cholesterol accumulation in astrocytes [60], which reminds us that the treatment of cholesterol-lowering drugs would be effective. In human patients with glaucoma, the use of statin (a cholesterol—lowering agent) is associated with a reduced risk of having glaucoma [165], suggesting that preventing cholesterol accumulation in astrocytes is also an attractive therapeutic target for glaucoma. Other studies have shown that connexin 43 (Cx43)-mediated astrocytic coupling is another therapeutic target for glaucoma [166]. Exposing human ONH astrocytes to high hydrostatic pressure causes internalization of Cx43 and decouples intercellular communication through gap junction [119]. Because connexin hemichannel is linked to inflammation, drugs maintaining the Cx43 gap junction (e.g., Dnegaptide) may be a candidate for astrocyte-targeted glaucoma treatment. A recent report has shown that overexpression of secreted phosphoprotein 1 (i.e., osteopontin) in astrocytes induces anti-inflammatory responses and protects RGCs [167]. In addition to blocking neurotoxic astrocytes, acceleration of neuroprotective astrocyte function may be an attractive candidate for glaucoma treatment.

## 8. Concluding Remarks

Astrocytes and tissue-specific astrocyte-lineage cells are widely distributed in the nervous system. Ocular astrocyte-lineage cells are essential for maintaining the physiological microenvironment around retinal neurons and optic nerves. Reactive changes and dysfunctions of these cell types are often found in human patients with ocular diseases such as glaucoma. At least a part of the pathological changes in glaucoma patients seems to be mediated by them. Many glaucoma-related molecules are expressed in astrocyte-lineage cells and cause their immune responses (Figure 5). As accumulating evidence suggests pathogenic roles of astrocytes in various neurodegenerative diseases, ocular astrocyte-lineage cells might be a novel therapeutic target for glaucoma in the near future. 

Transforming growth factor (TGFβ) and endothelin (ET-1) levels are elevated in serum and/or aqueous humor of glaucoma patients. These molecules up-regulate the expression of matrix metalloproteinases (MMPs) and extracellular matrix (ECM) proteins in the optic nerve head (ONH), which causes tissue remodeling. Purinergic signaling may be altered in glaucoma. Excess amounts of ATP could activate P2X7 receptors and induce inflammation, whereas activation of the P2Y_1_ receptor induces antioxidative responses that may suppress ONH remodeling. ONH astrocytes from glaucoma patients also show higher production of nitric oxide (NO) and tumor necrosis factor (TNFα), which may damage retinal ganglion cells (RGCs). Advanced glycation end-products (AGEs) are also increased in glaucoma patients’ eyes and can activate advanced glycation end-product receptors (RAGE) expressed in astrocyte-lineage cells and trigger inflammation. Many glaucoma risk genes are expressed in both astrocytes and Müller cells and contribute to their inflammatory responses. Molecules highly enriched in astrocyte-lineage cells may activate microglia and T cells to induce autoimmune glaucoma.

## Figures and Tables

**Figure 1 ijms-24-02747-f001:**
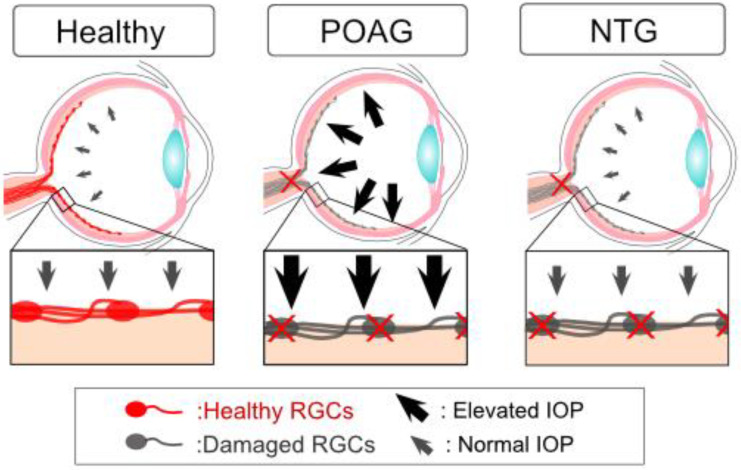
Elevated intraocular pressure (IOP) and glaucoma.

**Figure 2 ijms-24-02747-f002:**
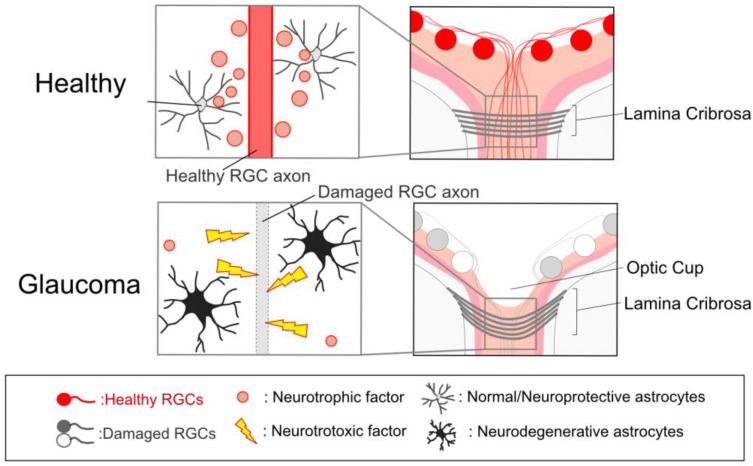
Cupping of the optic nerve head.

**Figure 3 ijms-24-02747-f003:**
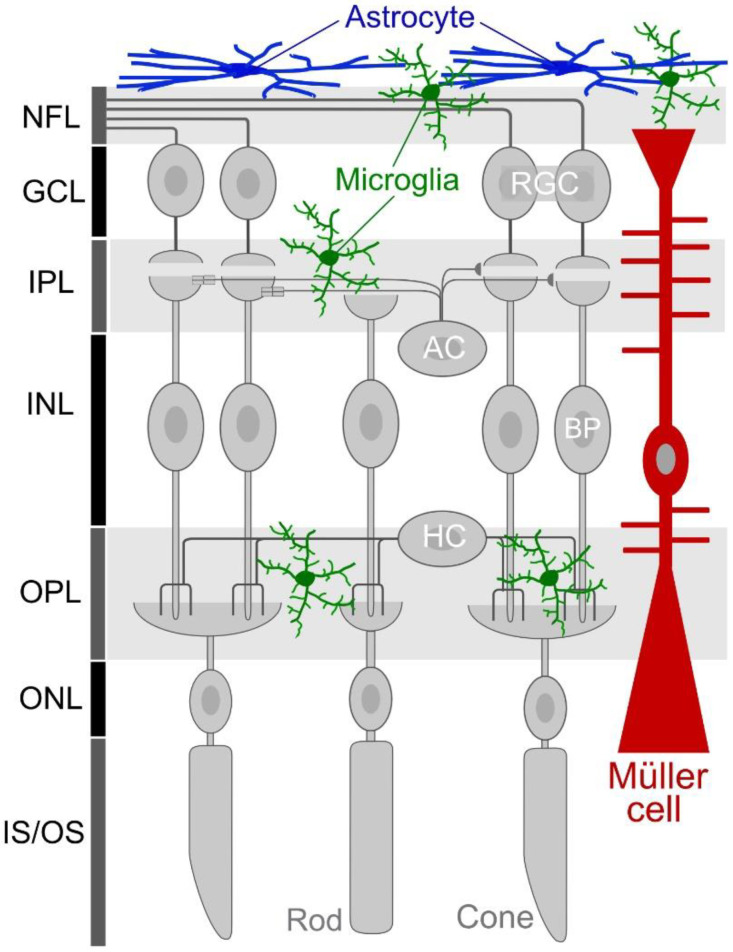
Glial cells in the retina.

**Figure 4 ijms-24-02747-f004:**
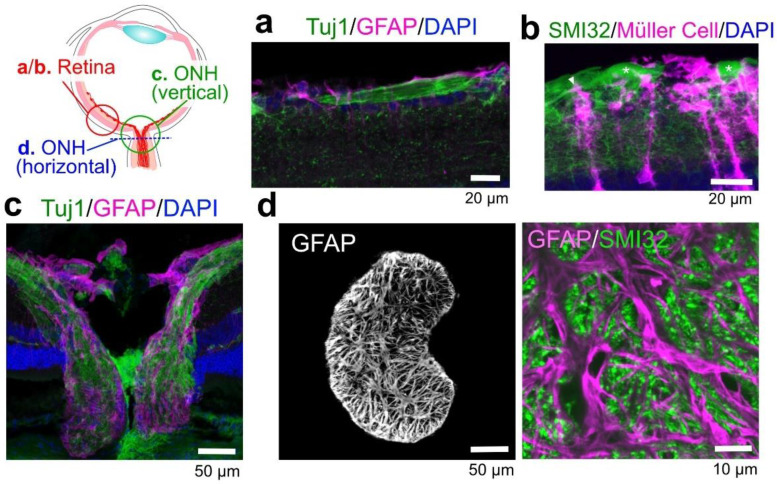
Astrocyte-lineage cells are closely associated with retinal ganglion cell (RGC) axons. (**a**) Retinal astrocytes (magenta) attach to Tuj1-positive RGC axons (green). (**b**) YC-Nano-expressing Müller cells [36] (magenta) elongate their processes toward and surround RGC soma (green, asterisk). A part of the Müller cell processes enwraps SMI32-positive RGC axons (arrow). (**c**) Vertical section of the optical nerve head (ONH). GFAP-positive astrocytes are highly enriched and closely attached to the optic nerve. (**d**) Horizontal section of the ONH: (**left**) an astrocyte forms a honeycomb structure and (**right**) RGC axons pass through the hole.

**Figure 5 ijms-24-02747-f005:**
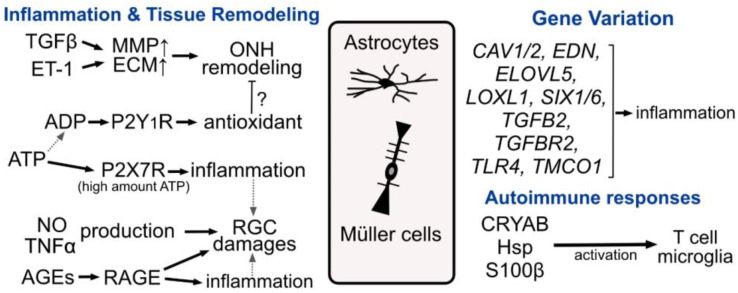
Glaucoma-related immune responses by astrocyte-lineage cells.

## Data Availability

Not applicable.

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
