# Peer review of "Astrocyte Immune Functions and Glaucoma"

_ijms, 2023, doi:10.3390/ijms24032747_

Round 1

Reviewer 1 Report

The review wrote by Shinozaki et al., takes into consideration the two astrocyte-lineage cells: astrocytes and Müller cells, in relation to ocular pathology like glaucoma, discussing the role of immune functions of astrocyte-lineage cells in the pathogenesis of such disease. The review is well written and it does present the important information about the astrocyte-lineage cells in the ocular milieu, especially when associated to a pathology like glaucoma. The subject of the review is not so completely new, but it allows the reader, especially the one who is new to the subject, to better understand it and retrieve the points that matter about such topic. The review can be accepted for publication in the present form with only a few minor points to adress reported here below:

-          The Authors have to pay attention to the template, in the first page, many things are missing. Please revise and modify accordingly.

-          The images should be placed immediately after their description in the paragraph. Please revise and modify accordingly.

-          Where is Table 1, which is cited in line 234? It is not present in the manuscript as well as in supplementary.

Author Response

 We sincerely appreciate reviewer #1 for his/her careful reading of the manuscript and constructive comments. We addressed all of the comments by revising the figures and text. Our responses to the reviewers’ suggestions are shown below (reviewers’ comments are in black and our responses are in blue).

-          The Authors have to pay attention to the template, in the first page, many things are missing. Please revise and modify accordingly.

As commented by reviewer 1, we used the template file and added all text and figures to it.

-          The images should be placed immediately after their description in the paragraph. Please revise and modify accordingly.

 We placed the images immediately after the paragraph in which the figure is referred.

-          Where is Table 1, which is cited in line 234? It is not present in the manuscript as well as in supplementary.

We thank reviewer 1 for pointing out the mistyping. We deleted it in the revised manuscript.

Reviewer 2 Report

The review article titled “Astrocyte immune functions and glaucoma” is a timely review of the pro-inflammatory roles of astrocytes in the progression of glaucoma. Overall manuscript is cogently and comprehensively written including diagrammatic presentations as easy references for the readers. The authors make a strong case that astrocyte cells play a crucial role in the pathogenesis of glaucoma similar to a parallel role in other neurodegenerative diseases. In addition, authors should discuss prospective studies exploring the role of astrocytes in the development of glaucoma.

Author Response

We sincerely appreciate reviewer #2 for his/her careful reading of the manuscript and constructive comments. We are pleased to hear that the reviewer found our review manuscript interesting. As suggested, we added new section 7 “Astrocytes as a therapeutic target for glaucoma” in which several examples of prospective studies show the role of astrocytes in the pathogenesis of glaucoma and their potential as a therapeutic target.

Reviewer 3 Report

Astrocyte immune functions and glaucoma.

The review article is very well written and will be very helpful for researchers to gain understanding of immune mechanisms involved in glaucoma

Introduction

Lines 37-42: Since the review article focuses on Glaucoma, not sure if these lines are necessary. The authors can integrate this better.

Line 37-38: only LPS is detrimental? The sentence seems to be very specific without many citations

Lines 67-68: ONH can be defined better.

Line 95-96: what do authors mean by optical fibers?

Figure 1 and Figure 2 can be improved and labelled better.

Author Response

 We sincerely appreciate reviewer #3 for his/her careful reading of the manuscript and constructive comments. We addressed all of the comments by revising the figures and text. Our responses to the reviewers’ suggestions are shown below (reviewers’ comments are in black and our responses are in blue).

Introduction

Lines 37-42: Since the review article focuses on Glaucoma, not sure if these lines are necessary. The authors can integrate this better.

We agree with this comment. The original version of the introduction is missing the link between astrocyte neurodegenerative/neuroprotective phenotypes and glaucoma. To introduce the readers to how astrocytic inflammatory responses (astrocyte immune functions) are attractive for the therapeutic target of glaucoma, we revised the text : (1) astrocytes are changeable into inflammatory (neurodegenerative) or anti-inflammatory (neuroprotective), (2) such neurodegenerative type of astrocytes can be seen in various neurodegenerative diseases, and (3) such phenotypes are modifiable and can be a therapeutic target.

Line 37-38: only LPS is detrimental? The sentence seems to be very specific without many citations

We added other examples of factors or conditions to induce neurodegenerative astrocytes including aging, human apoE4 knock-in, activated endothelial cells, and microglia-derived fragmented mitochondria.

Lines 67-68: ONH can be defined better.

We added the detailed description just behind the first appearance of the term ‘ONH’. To make it read easier, the description of the retina was moved to the end of the paragraph.

Line 95-96: what do authors mean by optical fibers?

As previously reported (Franze et al. PNAS2007), Müller cells have specific optical properties and orient the light direction to photoreceptor cells. Because they contribute to the transfer of light through the retina with minimal distortion and low loss, Müller cells seem to function like optical fibers. We did not add this information to the revised manuscript since it is out of our main topic.

Figure 1 and Figure 2 can be improved and labelled better.

In figure 1, we added the image for NTG to better understand IOP-independent glaucoma. Because labeling of icons made the figure busy, so we gathered them in the box under the images.

In figure 2, we inserted astrocyte images to show their phenotypes are associated with glaucoma pathology.